

# 1 Analyses of altimetry errors using Argo and GRACE data

2            J.-F. Legeais[1], P. Prandi[1], M. Ablain[1], S. Guinehut[1]

[1] Collecte Localisation Satellites, Parc Technologique du canal, 8-10 rue Hermès, 31520 Ramonville
Saint-Agne, France
*Correspondence to* : J.-F. Legeais (jlegeais@cls.fr)
**Abstract.**
This study presents the evaluation of the performances of satellite altimeter missions by comparing the altimeter
sea surface heights with in-situ dynamic heights derived from vertical temperature and salinity profiles measured
by Argo floats. This external assessment method contributes to altimeter Calibration and Validation analyses that
cover a wide range of activities. Among them, our approach focuses on the detection of altimeter drift and the
estimation of the impact of new altimeter standards that requires an independent reference. The methodology and
the Argo data used are first described and altimeter validation activities are then illustrated with some examples,
separating the analyses of the long-term evolution of the mean sea level and its variability, at global and regional
scales and results obtained via relative and absolute comparisons. The latter requires the use of the ocean mass
contribution to the sea level derived from GRACE measurements. Our analyses are related to different subjects
ranging from the estimation of the global mean sea level trend to the validation of multi-missions altimeter
products as well as the assessment of orbit solutions.
Even if this approach contributes to the altimeter quality assessment, the differences between two versions of
altimeter standards are getting smaller and smaller and it is thus more difficult to detect their impact. It is
therefore essential to characterize the errors of the method, which is illustrated with the results of sensitivity
analyses to different parameters. This provides an estimation of the robustness of the method and the
characterization of associated errors. The results also allow us to draw some recommendations to the Argo
community regarding the maintenance of the in-situ network.





## 1 Introduction

Since the early 1990s, several satellite missions have been equipped with altimeters allowing the estimation of Sea Level Anomalies (SLA) and the monitoring of the Mean Sea Level (MSL). This contributes to understand the role of the ocean in the Earth system and to assess the link with the global climate change. Altimeters are available onboard several missions currently on flight (Jason-2, SARAL/AltiKa, CryoSat-2, HY-2A) and providing no data anymore (TOPEX/Poseidon-T/P-, ERS-1&2, Jason-1, Envisat, Geosat Follow-On). Although sea level estimates are becoming more precise, there are still some uncertainties which can be distinguished at different temporal scales (long-term trend, inter annual signals and periodic signals) both at global and regional scales (Ablain et al., 2015). The major sources of errors are attributed to orbit solutions, instrumental corrections and some geophysical altimeter corrections such as the wet troposphere correction (Ablain et al., 2009, Couhert et al., 2014; Legeais et al., 2014; Rudenko et al., 2014).

Quality assessment of altimeter data can be performed thanks to internal comparisons (analyses of performances at crossovers points between ascending and descending tracks) and multi-mission cross calibration. A third approach is to compare with independent in-situ measurements. Tide gauges are commonly used (Mitchum 1998, 2000; Nerem et al. 2010; Arnault et al. 2011; Bonnefond et al. 2003, Valladeau et al., 2012) but even if they provide high temporal resolution measurements, the drawback is that only coastal areas are sampled and the instruments are not homogeneously distributed over the coasts (hemispheric bias).

In this study, we use Dynamic Height Anomalies (DHA) derived from the Temperature and Salinity (T/S) vertical profiles of the Argo network. The lagrangian profiling floats provide an almost global coverage of the open ocean with measurements from the surface to around 2,000 dbar for most of them and the objective of a global network of 3,000 operating floats has been achieved in 2007 (Roemmich and Team, 2009). The consistency between these in-situ measurements and altimeter SLA has already been discussed (Guinehut et al., 2006; Dhomps et al., 2011, Valladeau et al., 2012), showing that Argo DHA can be used as a reference (i) to detect drifts and jumps in the altimeter sea level time series to enable an assessment of the global and regional MSL trend and (ii) to assess the potential improvement provided by a new altimeter standard (e.g., orbit solution, geophysical corrections) in the altimeter SLA estimation. Argo data is thus a valuable tool to assess altimeter performances. However, the evolutions provided by the new algorithms allowing the sea level calculation (orbit solution, instrumental corrections, geophysical corrections, mean sea surface) become more and more difficult to assess (Stammer et al., 2014; Fernandes et al., 2015; Couhert et al., 2014). Hence, it is essential to determine to which extent the comparison with Argo independent measurements can be used to contribute to the quality assessment of these new algorithms and thus to better characterize the remaining errors of the method of comparison and its sensitivity to the various parameters. The paper is organized as follow: the different datasets used in our study are presented in section 2 and the details of the method of comparison of altimeter with in-situ measurements are given in section 3. Some examples of altimeter validation thanks to Argo data are presented in section 4 and section 5 is dedicated to the presentation of the sensitivity analyses of the method to various parameters. At last, concluding remarks are provided on the method uncertainty and the results also allow us to draw some recommendations for the Argo community regarding the maintenance of the in-situ network.

## 2 Datasets

### 2.1 Altimetry



Radar altimeters provide sea Surface height measurements which need to be referenced and corrected from
geophysical signals to determine SLA which can be compared with in-situ measurements. Along-track level 2
SSH from several satellite altimeters are used, where standards are updated compared with the geophysical Data
Record (GDR) altimeter products. Details of the SSH computation and time period for each altimeter are
available in the MSL part of the AVISO website (http://www.aviso.oceanobs.com/en/news/ocean-
indicators/mean-sea-level/processing-corrections/ ). Sea Level Anomalies (SLA) of all altimeter missions are
computed with a reference to the Mean Sea surface (MSS) CNES/CLS11 model (Schaeffer et al., 2012). Grids of
merged altimeter products (level 4) are also compared with in-situ data.
**2.2 Argo**
In this study, we use delayed mode and real time quality-controlled T/S profiles (Guinehut et al., 2009) from the
Coriolis Global Data Assembly Center (www.coriolis.eu.org). Following Roemmich and Gilson (2009),
considering a threshold of two thirds of the surface of the global open ocean covered by Argo floats, analyses
should be performed with in-situ data dating only from 2005 onwards. This is a relevant reference for the latest
altimeter missions (Envisat, Jason-1, Jason-2 and SARAL-AltiKa) and results in an in-situ dataset of more than
10,000 floats with about 900,000 T/S profiles distributed over almost the whole open ocean. Dynamic Height
Anomalies (DHA) are then computed from the integration of the vertical density profiles using a reference depth
and a synthetic mean dynamic height. The choice of the reference level is discussed in this paper.
**2.3 GRACE**
Altimeter measurements are representative of the total elevation of the sea surface (surface to bottom), that
includes barotropic and baroclinic components, whereas DHA from profiling floats are representative of the
steric elevation associated with the thermohaline expansion of the water column from the surface to the reference
level of integration (i.e. baroclinic component) (Dhomps et al., 2011). However, the relative comparison between
altimeter SLA and in-situ DHA may be sufficient to detect an anomaly between two different missions or the
impact of a new altimeter standard in the SLA calculation. The analysis of the absolute altimeter drift and bias
requires the addition of the mass contribution to the Argo dataset so that similar physical contents can be
compared. This ocean mass contribution is derived from the Gravity Recovery and Climate Experiment
(GRACE) satellite mission. It provides a series of Earth gravity fields in the form of truncated sets of spherical
harmonic (Stokes) coefficients at approximately monthly intervals (Tapley et al., 2004) whose temporal
variations can be used to estimate changes in the ocean mass distribution. In this study, two solutions are used.
The monthly grids of equivalent water height from the Groupe de Recherche en Geodesie Spatiale (GRGS
RL03; Biancale et al., 2014). When discussing the global altimeter performances, the temporal evolution of the
global mean ocean mass contribution from GRACE RL05 is also used, as proposed by the University of South
Florida – Satellite Oceanography Laboratory (available at:
http://xena.marine.usf.edu/~chambers/SatLab/Home.html, last access: July 9[th] 2014) and described in Johnson
and Chambers, 2013.
**3 Method**
The comparison of the altimeter SLA from a single mission is based on the along-track sea level measurements.
As the altimeter sampling is better than the in-situ coverage (offering a global coverage of the ocean –for Jasons
missions– versus a single T/S profile every ten days), grids of 10-day averaged along-track SLA are interpolated
for each altimeter mission at the location and time of each T/S profile (bi-linearly in space and linearly in time).





Similarly, the quality assessment of gridded merged SLA altimeter products (L4) can be estimated after
collocation with the in-situ profiles.
In addition, the in-situ DHA are referenced to a synthetic mean Argo dynamic height calculated over the period
2003 to 2014. It is critical that altimeter SLA is compared relative to the same temporal reference. It affects the
global correlation and the regional trend differences between both types of data (see example in the paper). This
is performed by removing the mean of AVISO SSALTO/DUACS SLA maps for 2003-2014 (AVISO Handbook,
2014) from each altimeter measurements.
In order to improve the correlation between both types of data (and thus increase our confidence in the results),
outliers (corresponding to differences between altimeter SLA and in-situ DHA greater than 0.20 m) are filtered
out. All associated measurements are located in regions of high ocean variability, indicating that our method of
collocation leads to an increased error of the results in these regions. This validation step contributes to reduce
this error and improves the accuracy of the method. Global and regional statistics on the sea level differences are
then generated and various diagnoses are produced from these statistics in order to detect potential anomalies in
altimeter data.
For global analyses (trends, inter-annual and annual signals), an alternative method of comparison consists in
computing global mean time series of altimeter SLA and Argo DHA with the same temporal sampling and then
subtract the time series. This approach is discussed further in the paper.
**4 Altimeter Sea Level Validation**
In this section, the usefulness of the altimeter comparison with Argo floats is described with some examples. For
each of them, different spatial and temporal scales are addressed among the following via relative or absolute
comparisons (without or with the ocean mass contribution): the long-term evolution of the mean sea level or its
variability at global or regional scales.
**4.1 Detection of global altimeter drifts**
At global scale, the MSL trends of the Envisat and Jason-1 missions differ by 1.0 mm/yr over the period 2004-
2011 (Prandi et al., 2013). The absolute comparison of both altimeter MSLs with Argo and GRACE
measurements indicates that the MSL drift is greater for the Envisat than Jason-1 mission with a 1.4 mm/yr
difference (Fig. 1). The altimeter comparison with tide gauges measurements over the same period highlights a
0.9 mm/yr difference (Prandi et al., 2013) which confirms the greater drift of the Envisat mission. Thus the
combination of different types of in-situ data allow to detect and identify the origin of global altimeter MSL
trend discrepancy between two missions that cannot be addressed by internal comparison only. This Envisat drift
is well known and has been related with the altimeter standards and instrumental corrections used for the
estimation of the Envisat sea level (Ollivier et al., 2012). This is no more observed with the use of the Envisat
reprocessed measurements which have made both altimeter trends more homogeneous.
**4.2 Detection of the impact of new altimeter standards**
The Argo steric heights are used as a reference in order to estimate the impact of new altimeter standards used
for the altimeter sea level calculation. For instance, the use of the GDR-D orbit solution leads to a regional
East/West hemispheric bias in the spatial distribution of the Jason-1 MSL trends (Legeais et al., 2015). As Argo
measurements are considered to be free of this regional anomaly, the relative comparison of the MSL trends
differences between SLA and DHA (computed in two different East/West regions where the greatest differences
are observed) illustrate the strong regional discrepancy obtained with the GDR-D orbit solution (Figure 2a: 2.3





mm/yr). The in-situ Argo network is used to assess the impact of the updated GDR-E orbit standard in the Jason-
1 MSL calculation. The significant reduction of the hemispheric trend differences (Figure 2b, right: 0.1 mm/yr)
proves that the estimation of the altimeter SLA is improved with this new altimeter standard since the regional
discrepancies of the MSL trends are reduced. As discussed in Valladeau et al, 2012, the global Argo
measurements are the only in-situ external reference that allows us to discriminate such an impact regarding the
altimeter MSL.
**4.3 Detection of the impact of new altimeter products**
The independent Argo sea level estimations can also be used at global scale to distinguish two different altimeter
L4 merged products by relative comparison in terms of MSL variability. The Sea Level Climate Change
Initiative (SL_cci) project has provided climate-oriented Sea Level products (Cazenave et al., 2014; Ablain et al.,
2015) and we are interested in characterizing the differences between the SL_cci v1.1 ECV product and the 2014
SSALTO/DUACS time series (AVISO Handbook, 2014; Pujol et al., 2015). In order to isolate specific signals
and better discriminate the datasets, different frequencies of the differences between altimeter SLA and in-situ
DHA are distinguished. The correlation and the standard deviation of these differences are estimated over the
global ocean at different temporal scales. This is illustrated on Figure 3 using the SL_cci (triangles) and
SSALTO/DUACS 2014 (circles) products, thanks to the Taylor diagram formalism (Taylor, 2001). Such
diagram provides a way of graphically summarizing how closely different patterns match observations (in-situ
data: gray dot on the bottom axis). The similarity between two patterns is quantified in terms of their correlation,
their centered root-mean-square difference and the ratio of their variances. The statistics are indicated for the
total signals (in black) but also for the annual cycle (in green), high frequencies (in red) and inter annual signals
(in blue). The very high correlation (0.98) found between altimetry and in-situ data for the annual cycle only (in
green) indicates that this signal is at the origin of most of the similarities between both types of data, showing
that it is necessary to remove these annual variations before analyzing other frequencies. This Taylor diagram
reveals that both altimeter products cannot be significantly distinguished regarding the total signals (in black),
their annual cycle (in green) and their high frequencies (in red). At low frequencies (in blue), the SL_cci product
(triangle) is more in agreement with in-situ data than the SSALTO/DUACS product (circle) which is in favor of
a product dedicated to climate studies. However, the correlations of each altimeter data with the in-situ reference
are similar.
Furthermore, the validation of the reprocessed AVISO/DUACS 2014 products (AVISO Handbook, 2014) has
shown that the differences with the previous release of this product (AVISO/DUACS 2010 reprocessing) are
sometimes reduced for some statistics (Pujol et al., 2015). The characterization of the differences between these
products by relative comparison with Argo data at regional scale in terms of variance differences between SLA
and DHA is an additional illustration of the asset of this independent in-situ reference. Figure 4 indicates that in
the Bay of Bengal, the variability of the altimeter SLA minus in-situ DHA differences is reduced (-1 cm$^2$) with
the use of the new altimeter release. The statistics in this area (Table 1) indicates that the reprocessed altimeter
dataset provides a slightly greater correlation and a reduced rms of the differences with the in-situ reference.
This indicates that the Argo in-situ measurements can be used to assess the impact of a new altimeter product at
regional scales even in a small area.
**5 Sensitivity of the method**




This section focuses on the determination of the errors of the method of comparison of altimetry with in-situ
Argo data and provides sensitivity analyses of the method to different parameters. For each analysis, the impact
of a parameter is estimated regarding the long-term evolution of the mean sea level or its variability at global or
regional scales. In the following, the term "error" is considered as a quantity that would be removed if it was
known whereas the term "uncertainty" is associated with the confidence that can be attributed to the estimation
of a given parameter.
**5.1 Format of altimeter data**
As presented earlier, the assessment of a single altimeter missions is based on the collocation of each in-situ
profile (linearly in space and time) with grids of 10-days box-averaged along-track SLA with boxes of 1° latitude
x 3° longitude in order to take into account the number of altimeter tracks per cycle and also the rather zonal
ocean circulation because of the Coriolis force associated with the rotating effect of the Earth. The sensitivity of
the method to this size of boxes is estimated by comparing the results with 1°x1° grids of along-track altimeter
SLA. The amplitude and phase of the annual signal of the SLA – DHA differences are not affected by this
change of box size, neither the trend of the differences (not shown).
The variance of the SLA-DHA differences is computed for the time series of each Argo floats, using
successively the two different sizes of boxes for altimetry. The histogram of the difference of these variances for
all Argo floats (Figure 5) provides a mean of +1,3 cm$^2$, which indicates that averaging along-track altimeter data
with 1°x3° boxes makes altimeter data more coherent with in-situ Argo observations. This processing is
therefore chosen for the comparisons.
**5.2 Error of collocation**
The variability of the SLA – DHA differences are larger in regions of high ocean variability since the collocation
of altimeter and in-situ measurements is performed by interpolation of 10 days box-averaged along-track SLA at
the position and time of each Argo profile. Hence, the time of two co-located altimeter and in-situ measurements
may not be strictly the same and the associated impact may be higher in areas of high ocean variability where the
ocean state may change significantly within less than 10 days. Note that this effect could be reduced by
computing maps of altimeter measurements by optimal interpolation. However, this is very time consuming
since a set of grids has to be computed for a specific mission as soon as the impact of a new altimeter standard
has to be evaluated.
In order to estimate the error of the method associated with these regions of high ocean variability, the
comparison of altimeter data with Argo measurements could be performed after removing areas where the ocean
variability is higher than a given threshold. In terms of spatial coverage, the lower this threshold, the larger areas
are removed. The detection of altimeter drift is not affected by the exclusion of areas of high ocean variability.
Indeed, the 2.07 mm/yr trend of the mean differences between SSALTO/DUACS and Argo DHA (900 dbar
reference) is not significantly changed when areas of ocean variability higher than 100 cm$^2$ are excluded (2.16
mm/yr). This will be confirmed with results described later in this paper regarding the sensitivity to the spatial
sampling of the Argo network. Figure 6 (left) illustrates that the lower the threshold on the ocean variability, the
larger areas are removed and thus, a lower number of observations is available. The right panel indicates that
when larger areas are removed, the correlation between altimeter SLA and Argo DHA gets lower and the rms of
the differences (expressed in percentage of the altimeter variance) increases. This suggests that the areas of large
ocean variability significantly contribute to the global statistics computed between altimetry and Argo data.



However, this does not allow us to determine whether an increased sampling of these regions by the Argo
network would improve the results of altimetry validation.
In addition, our study focuses on the altimeter quality assessment. In particular, the estimation of the global
altimeter MSL drift is not considered to be significantly affected by the fact that some regions of the ocean are
not covered by the Argo network (e.g. the Indonesian throughflow, the Gulf of Mexico). The steric contributions
of such regions may be of importance for sea level closure budget studies (Dieng et al., 2015b), but similarly
with comparisons to tide gauges, they do not prevent from estimating the global MSL evolution.

**5.3 Impact of the temporal reference period**

When comparing both types of data, altimeter SLA and in-situ DHA should have similar physical contents and
in particular the same inter annual temporal reference. This does not affect the global trend differences but it
directly impacts the trend differences at regional scales. In addition, the detection of the evolution provided by a
new altimeter standard or product in terms of global correlation between all collocated altimeter SLA and in-situ
DHA may be distorted whether the temporal reference is homogeneous or not between both types of data. Table
2 indicates that without a homogeneous temporal reference, the reprocessed AVISO SSALTO/DUACS DT 2014
product is more correlated with Argo DHA than the previous release of these products. However, no difference
of correlation is observed when the anomalies are computed with the same temporal reference. This illustrates a
particular type of error of the method of comparison (different temporal references) that can be corrected (by
referencing both datasets on the same period).

**5.4 Impact of the GRACE data set and associated errors**

At regional scales and particularly in the tropical ocean, total altimeter and steric annual signals are in phase
(Dhomps et al., 2011, Legeais et al., 2015) but due to the spatial distribution of the ocean on the Earth and
seasonal hemispheric signals, the global time series are affected by a quadratic phase shift (Figure 7 and Chen et
al., 1998). Regarding the ocean mass contribution to the sea level, its annual signal has a larger magnitude
(twice) than total and steric signals and is in phase with the total altimeter global MSL. The addition of the mass
contribution from GRACE to the Argo dataset provides homogeneous physical content with altimeter SLA
(except the deep steric contribution) (Figure 7), which is required to estimate the altimeter absolute drift. In
addition, Figure 8 highlights that the amplitude of the annual signal of the global differences between the total
altimeter signal and the steric DHA is about 10 mm (in red) and it is significantly reduced when the ocean mass
contribution is also withdrawn (in blue). This demonstrates the relevance of this ocean mass contribution for the
detection of the altimeter absolute drift detection.
The analysis of altimeter absolute drift requires a good accuracy of the long term changes in ocean mass (trends,
inter-annual to decadal variations) and two important corrections have to be taken into account for such analyses.
The first one is the Glacial Isostatic Adjustment (GIA) which is a gravity effect. It is related to the Post Glacial
Rebound (Tamisiea and Mitrovica, 2011) whose oceanographers are not interested in since they rather want to
assess the current mass movements. The GRACE ocean measurements have to be corrected of a GIA of 1.1
mm/yr (Chambers et al., 2010). However, GIA does not represent the mass redistribution of continental ice to the
oceans, which should be corrected. Based on tests with different ice loading histories and Earth models, the GIA
uncertainty is estimated to be 30% (~0.3 mm/yr) (Chambers et al., 2010). The second essential ocean mass
correction deals with the degree 1 geocenter motion. Satellites move about the mass center of Earth but it moves
over time relative to the fixed geometric center and we are interested in the mass loss relative to a fixed frame



(i.e., the crust). In addition, the redistributions of ice from Greenland, Antarctica, and mountain glaciers affect
geocenter trends and although the effects offset somewhat, the uncertainty associated with this correction of
geocenter motion in terms of equivalent sea level is estimated to be 0.1 mm/yr (Swenson et al., 2008; Chambers
et al. 2007). In addition of these GIA (0.3 mm/yr) and geocenter (0.1 mm/yr) uncertainties, the global mean
ocean mass evolution is also affected by the harmonic coefficients fit uncertainty (0.1 mm/yr) and the leakage
from land to the ocean. This latter effect can be taken into account by removing a 300 km coastal band but the
remaining uncertainty is also of order of 0.1 mm/yr. The detection of the altimeter absolute drift is thus
significantly affected when introducing GRACE measurements.
Regarding the global altimeter drift, Figure 9 displays the temporal evolution of the differences between
altimetry and the sum of Argo DHA plus GRACE measurements using the grids of equivalent sea level (GRGS
solution, Biancale et al., 2014) and the global mean ocean mass (Johnson and Chambers, 2013). A 1 mm/yr
difference is observed between the altimeter drift estimated with the former (0.8 mm/yr) and the latter (-0.21
mm/yr) dataset. At inter annual scale, opposite temporal variations between both time series can be observed of
the order of several millimeters (such as during year 2008). These discrepancies are attributed to the difference
of processing of these datasets: the spherical harmonic coefficients are addressed differently (in particular the
degree 0 and 1 coefficients) and the ocean mass time series obtained with the GRGS dataset has been adjusted
for a -1.1 mm/yr GIA effect whereas this effect is already taken into account in the global mean ocean mass time
series. In addition, the so-called leakage of the continental signal over the oceans is not treated the same way.
Note that the method of comparison also contributes to the observed discrepancies (GRGS solution collocated to
Argo profiles versus global mean difference) but it is not believed to be a first order contribution to the error.
This illustrates that all the uncertainties mentioned above can significantly affect the estimation of the altimeter
absolute drift.
**5.5 Impact of the temporal sampling of the Argo floats**
The Argo floats provide vertical T/S profiles every 10 days. This is a good compromise in order to sample the
ocean variability and to ensure a long enough life time of the floats. For comparison, altimeter missions such as
Jason missions provide a global coverage of the ocean within the same period. The validation of altimeter
measurements by comparison with the in-situ profiles may be affected by a different temporal sampling of the
Argo floats. With a full sampling of the in-situ network, an East/West hemispheric bias of the regional MSL
trends is observed when computing the trend of the differences between altimeter Jason-1 SLA and in-situ DHA
in each hemisphere (Figure 10). The difference of trends between each area is of -1.38 mm/yr over mid 2004-
2010 with the GDR-C orbit solution (Fig. 10a) whereas it is reduced to -0.13 mm/yr with the GDR-D orbit
solution (Fig. 10b). This indicates that this updated altimeter standard improves the regional homogeneity of the
altimeter SLA but given the uncertainty associated with these trend estimations (more than 0.5 mm/yr over this
period), these results are close to the limit where both these values can be distinguished with enough confidence
in the results.
The goal is to assess whether this result is affected by a change the temporal sampling of the Argo floats. The
trend of the differences between the altimeter SLA and in-situ DHA is computed as before for each hemisphere
with both altimeter standards but only one out of three in-situ profiles is used which leads to a monthly sampling
for all floats instead of 10 days. The East/West hemispheric trend differences become -0.98 mm/yr and 0.67
mm/yr with the GDR-C and GDR-D standards respectively. This means that in these conditions, none of the





standards allow the reduction of the hemispheric discrepancies with respect to the in-situ independent reference.
This kind of analysis of impact of a new altimeter standard is thus sensitive to the sampling frequency of in-situ
floats.
**5.6 Impact of the spatial sampling of the Argo network**
The target of a network of 3000 Argo floats has been achieved in 2007 and they now provide an almost global
coverage of the open ocean. This targeted number of floats has not been determined in order to allow altimetry
validation in particular. The impact of a reduced spatial coverage of the network on the altimetry validation is
analyzed in terms of regional coverage, trends of the differences and coherence between both measurements.
Different selections of the floats have been performed and Figure 11a displays the number of valid profiles over
2005-2012 from all Argo floats whereas the Figure 11b shows the number of valid profiles when only 25% of
the floats are used (selected in the list of instruments following the increasing order of their WMO number).
With this selection, the spatial coverage is strongly affected and some regions are not sampled at all over the
period.
Focusing on the altimeter drift detection and in spite of this reduced spatial coverage, the global trend of the
differences between altimetry and Argo steric heights are not significantly modified (within 0.04 mm/yr) when
different sub samplings of the network are used (50% or 25% of the number of instruments). This is in
agreement with the lack of impact of the high ocean variability areas on the global altimeter trend estimation, as
described earlier. In order to have a consistent approach, the same sensitivity test has been performed as the one
used for the impact of the temporal sampling (see previous section). The trends of the differences between the
altimeter SLA and in-situ DHA are computed separating the eastern and western hemispheres using both Jason-1
altimeter standards but only 50% of the Argo floats are used in the comparisons. The East/West hemispheric
trend differences are -1.2 mm/yr and -0.1 mm/yr with the GDR-C and GDR-D altimeter standards respectively,
which are very similar to the differences obtained with all floats (-1.4 mm/yr and -0.1 mm/yr respectively). This
suggests that the reduction of the number of floats (and thus of the spatial coverage) has also no significant
impact on the detection of altimeter drifts at regional scale.
In addition, Figure 12 shows the Taylor diagram (Taylor, 2001) between AVISO SSALTO/DUACS altimeter
merged products and the Argo in-situ steric heights (with the addition of the GRACE GRGS ocean mass
contribution) with different sub sampling of the Argo network. The performance obtained with 25% of the floats
appears to be slightly deteriorated but the different points are very close to each other and as above for the global
and regional trends, this confirms that the validation of altimeter measurements is little affected by a reduction of
the number of Argo floats and a reduced spatial coverage of the in-situ network.
The reduction of the temporal and spatial sampling of the Argo floats could have been considered to have similar
effects but the same sensitivity analyses have been performed (impact of Jason-1 altimeter standards on the
regional hemispheric trend discrepancies) leading to opposite conclusions regarding the sea level trends (impact
versus no impact). This indicates that according to the method of sub sampling, the distribution of the in-situ
information (in space and time) are statistically different, leading to a different impact on the altimeter sea level
estimation. This will be further illustrated in the following section.
**5.7 Reference depth of Argo profiles**
The integration of the Argo T/S profiles for the computation of the in-situ steric dynamic heights requires a
reference level (pressure) and the deeper the reference level, the more information from the T/S profiles is taken



into account through the water column but the more T/S profiles are not used (those who don't reach the
reference level). Thus, we first aim at determining the impacts of a given reference depth of integration on the
global and regional Argo spatial sampling, on the estimation of the global MSL trend and in terms of sea level
variance.
**5.7.1 Impact on the global and regional coverage**
According to the reference pressure used to integrate the in-situ density profiles, no DHA will be computed for
all the floats whose mean maximum pressure does not reach this reference level. At global scale, only 6% of the
floats are missed with a reference level at 900 dbar but this proportion increases to 29% at 1400 dbar and 52% at
1900 dbar.
At regional scale, the floats used with a 900 dbar reference pressure provide a very homogeneous ocean
coverage (Figure 13a) and associated discarded floats whose reference pressure is shallower are mainly located
in the Pacific western boundary current, in the Mediterranean Sea and a few are found in the tropical Atlantic
and Eastern Pacific Ocean (Figure 13c). The map of the discarded floats with a deep reference level (1900 dbar)
(Figure 13d) indicates that floats with a mean max depth between 900 dbar and 1400 dbar (in light blue and
green) are mainly located at equatorial latitudes of all ocean basins. In these areas, the water column is very
stratified and the steric signal is thus confined in the upper layer. Floats reaching depths between 1400 and 1900
dbar (in orange and light red) are mainly found at subpolar latitudes where signals are more barotropic compared
to lower latitudes (Luyten et al., 1983). Floats reaching depths deeper than 1900 dbar are relatively well spread
out over the ocean with increasing density in the western boundary currents of the north hemisphere. Thus, with
a deep reference depth, the water column will be better sampled over the global ocean (which improves the
retrieved steric signal) but we will miss a significant part of this steric signal, especially at equatorial latitudes.
This illustrates the balance to be found between the horizontal (shallow reference level) and vertical (deep
reference level) sampling of Argo floats.
**5.7.2 Impact on the global MSL trend estimation**
An estimation of the global altimeter absolute drift is provided by the global mean sea level differences between
altimetry and the sum of Argo steric heights with the GRACE ocean mass contribution. This is illustrated on
Figure 14 with various subsets of DHA derived from the Argo network, allowing the distinction of the effect of
the horizontal and vertical sampling of the ocean by the floats. The altimeter drift estimated with all DHA from
900 dbar profiles (in red) is of 1.5 mm/yr. Among these profiles, the selection of those whose maximum depth is
at least 1900 dbar (impact of the horizontal sampling) has no impact in terms of global correlation between
altimetry and Argo measurements (0.84 in both cases). There is a relatively low impact (-0.2 mm/yr) on the
altimeter drift which is reduced to 1.3 mm/yr over the period (in blue). The use of all DHA from 1900 dbar
profiles leads to an improved correlation between altimetry and in-situ data (0.87) and the impact of this
increased vertical sampling on the altimeter drift detection (in green) is greater than previously (-0.4 mm/yr) and
leads to a 0.9 mm/yr drift. Therefore, the choice of a deep reference level for Argo DHA provides a better
estimation of the baroclinic signal (improved vertical sampling) which is more in agreement with the observed
signal by altimetry. This is in favor of an improved estimation of the absolute altimeter drift detection.
The use of a deep versus shallow reference level turns out to be equivalent to a reduction of the ocean coverage
by Argo floats (horizontal sampling). As previously discussed with the analysis of the sensitivity to the temporal
and spatial sampling of the floats, this kind of sub sampling associated with the reference level affects the





estimation of the global absolute altimeter sea level trend. The 0.6 mm/yr total difference observed between the
shallow and deep reference levels on Figure 14 is an estimation of one of the contributors to the error of the
method of comparison.

**5.7.3 Impact in terms of variance: altimetry multi vs mono mission**

We now describe two examples at global and regional scales illustrating that the comparisons of altimeter
measurements with Argo in-situ data in terms of variance are affected according to the reference level of
integration of steric heights. At global scale, the Taylor diagram of Figure 15 presents the correlation and the
standard deviation of the differences between altimeter multi-missions merged SLA and the Argo steric DHA.
With a deep reference level (1900 dbar), the altimeter (grey circle) and in-situ (black circle) time series have the
same standard deviation whereas a reduced variability is found with the in-situ steric measurements referenced to
a shallower level (900 dbar) with a 0.85 proportion compared with altimeter SLA. In addition, the correlation
between both types of data is also deteriorated. This has to be taken into account when assessing the impact of a
new altimeter standard or new product for instance.
At regional scales, Dhomps et al. (2011) reveal that the correlation and the regression coefficients between SLA
and DHA vary spatially with a latitude dependency at the first order. In particular, their Fig. 5 suggests that the
Southern Ocean is the place where the water column has to be sampled at the deepest level to estimate the steric
signal. At high latitudes, the baroclinic signal below 1000 m depth significantly improves the correlation
between SLA and DHA, the sea level variability being largely influenced by the deep baroclinic signals. We
illustrate this with Figure 16 which indicates that the variances of the differences between altimeter SLA and in-
situ DHA are different whether the altimeter SLA is derived from mono mission (TOPEX, Jason-1 & 2) or
multi-missions grids of SLA. In particular, with DHA referenced to 900 dbar (left panel), adding missions
reduces the altimeter / Argo consistency in the high ocean variability areas of the Antarctic Circumpolar Current
(ACC) (blue, negative values of -5 cm$^2$ on average). On the other hand, this tendency almost disappears in the
ACC with the use of DHA referenced to 1900 dbar (right panel). This result is explained by the difference of
variance of the water column as seen by altimetry or in-situ data in this region. Figure 17 indicates that the
variance of mono mission and multi missions altimeter products (collocated to Argo profiles) are very close to
each other in the ACC but the variance of the Argo steric heights referenced at 900 dbar is significantly lower.
Thus with this reference level, both altimeter products cannot be distinguished by comparison with Argo data.
With a 1900 dbar reference level, the variance of the Argo steric heights becomes similar to the values obtained
with altimeter products in the ACC and the Argo measurements become relevant for the quality assessment of
the altimeter products. This illustrates that according to the ocean characteristics, the analysis of the variance of
the water column and thus the differences between altimetry and Argo measurements are highly sensitive to the
reference depth of integration of the Argo profiles.

**5.8 Impact of the deep steric contribution**

In addition of the sensitivity to the reference depth of integration of Argo density profiles (as described in the
previous section), the estimation of the altimeter drift is also affected by the deep steric contribution (deeper than
the reference level of Argo floats) which is not taken into account in our approach. This contribution has been
extensively discussed in the recent years since the heat uptake in the deep ocean is suspected to explain the pause
in the global mean air and sea surface temperature evolution observed since the early 2000s (e. g. Trenberth and
Fasullo 2013; Watanabe et al. 2013; England et al. 2014). Comparing altimeter SLA with the sum of the steric



signal and the ocean mass contribution, Dieng et al., 2015a estimate the deep steric contribution (deeper than
1500 m) to be $0.3 \pm 0.6$ mm/yr and $0.55 \pm 0.6$ mm/yr over the period 2005-2012 and 2003-2012 respectively.
Llovel et al. (2014) provide an estimation of $0.0 \pm 0.7$ mm/yr over the former period. The associated
uncertainties include the formal error adjustment and the systematic errors associated with the observing system.
The problem with the estimation of the deep steric contribution is that it requires the knowledge of the steric
contribution from the upper ocean and the comparison of different global steric sea level datasets indicates that a
significant uncertainty remains on this estimation (Dieng et al., 2015a). This suggests that for the moment, there
are still too large errors associated with the estimation of the deep steric contribution to detect absolute altimeter
sea level drift with regards to climate users requirements: 0.3 mm/yr over 10-year (GCOS 2011). Note that some
deep profiling floats (about 4000 m) have been recently launched in the context of the Euro-Argo Improvements
for Marine Services (E-AIMS, 2013) which should help to better characterize the deep steric contributions and
assess their impact on the altimeter quality assessment. As an illustration, Figure 18 display the time series of the
DHA derived from the profiles of such a float drifting off the Bay of Biscay (WMO 6901632) with different
reference levels of integration varying from 900 dbar down to 4000 dbar together with the collocated altimeter
SLA (in brown). A very good coherence is globally found between all curves. A 3 cm bias is observed between
DHA 900 dbar and DHA 1900 dbar but also between DHA 1900 dbar and DHA 3400 dbar. The steric signal
deeper than this pressure seems to be much reduced since almost no bias is observed between 3400 dbar and
4000 dbar. In addition, the correlation between SLA and DHA significantly increases from 900 dbar (0,70) to
1900 dbar (0,90) and reaches up to 0,92 at 3400 dbar. Thus, the use of deep reference levels increases the
coherence between the in-situ and altimeter sea level estimations but regarding the altimeter drift detection, it is
fundamental to have enough in-situ measurements over a long period so that the in-situ sea level trend can be
used as a reference with enough confidence and is really representative of the global ocean.
**6 Conclusions**
The internal consistency check and the comparison with other altimeter missions cannot systematically provide
enough information for the quality assessment of altimeter sea level measurements. The in-situ dynamic heights
derived from the Argo network can be used as an independent reference for the analysis of the relative mean sea
level temporal evolution (including the detection of global and regional MSL drift and anomalies) but also for
the detection of the impact of new altimeter standards or products used to calculate the sea surface heights. Our
method constitutes an essential approach which has a strong synergy with results derived from the altimetry
comparison with tide gauges since the confrontation of both methods improves the confidence in the results. We
have demonstrated that it is possible to detect altimeter drifts at global and regional scales and to characterize the
impact of new altimeter standards. However, the improvements provided by these new standards and products
become more and more reduced and the searched differences may be hidden by the errors of the method. It is
thus necessary to better characterize the capacity of the method to distinguish the performances of two altimeter
products. Hence, this study focuses on the sensitivity of the altimeter / in-situ sea level comparisons to different
processing parameters.
The estimation of the absolute altimeter mean sea level drift requires the additional information related to the
mass contribution to the sea level that can be derived from GRACE satellite measurements. We have shown that
there is a strong sensitivity to the different datasets available. In addition, regarding the long term trend of the
global MSL, there are significant uncertainties associated with the GIA correction (0.3 mm/yr), the geocenter


motion (0.1 mm/yr), the fit of the harmonic coefficients (0.1 mm/yr) and the leakage from land to the ocean (0.1
mm/yr). The estimation of the altimeter MSL trend is thus directly affected by these uncertainties related with
the use of GRACE measurements.
Sensitivity analyses performed on the Argo network have indicated that the spatial coverage of the ocean
sampled by the instruments is significantly reduced as soon as a limited number of floats are used in the
comparisons. However, this hardly affects the global correlation between altimeter SLA and the in-situ DHA
plus mass contribution, neither the variance nor the trend of their differences. In addition, the 10-day temporal
sampling of Argo floats was not designed for satellite altimetry validation purposes. We have shown that a
reduced temporal sampling of the floats can prevent us from detecting the impact of a new altimeter standard.
The same diagnosis has been used to assess the impact of the reduction of the temporal and spatial sampling of
Argo floats, leading to opposite conclusions. This suggests that the resulting distributions of the in-situ profiles
(in space and time) are different, leading to a different impact on the regional sea level trend estimation.
The choice of the reference level of integration of the Argo T/S profiles for the computation of the steric
dynamic heights directly affects the global and regional coverage of the ocean by Argo floats. A relatively
deeper reference level can be assimilated to an additional sub sampling effect since it allows a better vertical
sampling of the water column (more in agreement with what is seen by altimetry) but this leads to a reduced
horizontal sampling of the ocean; the impact of the former being more than twice compared with the latter in
terms of altimeter MSL trends estimation over a 8 years period. In some regions such as the Southern Ocean, the
comparison with the altimeter sea level requires a deep reference depth so that the variance content of the water
column is similar between altimetry and in-situ data.
Considering all the sources of errors discussed in this study including the method of collocation, the impact of
the reference depth of Argo profiles, the uncertainty on GRACE ocean mass datasets and the error estimation on
the deep steric contribution, this suggests that the uncertainty associated with the obtained altimeter drifts is at
least of the order of 1.0 mm/yr. The future evolution of the Argo network such as the deployment of deep Argo
floats (4,000m) should contribute to improve the results and our approach will be an asset for the quality
assessment of new altimeter missions such as Sentinel-3, Jason-3 and SWOT.
Following the results of this study, the Argo community should be supported to maintain and improve the
deployment of Argo profiling floats. In particular, the temporal sampling of the Argo floats should be maintained
with at least the existing temporal coverage and the vertical extension of the Argo profiles should be extended to
deeper levels. In addition of these recommendations, enlarged network coverage at high latitudes and over
shallow waters, as well as an improved quality control of the data would also contribute to improve the altimeter
quality assessment thanks to the Argo network.
**7 Acknowledgements**
This work was supported by the CNES thanks to the SALP project and was partly carried out as part of the FP7-
SPACE E-AIMS project - grant agreement 312642.



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



| Argo DHA 1900 dbar | Correlation | rms of the differences (cm) |
|---|---|---|
| SSALTO/DUACS DT 2010 | 0.89 | 3.94 |
| SSALTO/DUACS DT 2014 | 0.90 | 3.76 |

1    Table 1 : Statistics between altimeter products and in-situ DHA with an homogeneous reference period of the

2    altimeter SLA and in-situ DHA (2003-2011) in the Bay of Bengal (-5°S/+20°N; 80°E/95°E); Argo DHA are

3    referenced to 1900 dbar.



| Global correlation | Non homogeneous temporal reference | Homogeneous temporal reference |
|---|---|---|
| AVISO SSALTO/DUACS 2010 | 0.87 | 0.90 |
| AVISO SSALTO/DUACS 2014 | 0.90 | 0.90 |

1   Table 2 : Global correlation between all collocated altimeter SLA (AVISO SSALTO/DUACS 2010 and 2014)

2   and in-situ DHA from Argo profiles (with a reference depth of 1900 dbar and a 2003-2011 temporal reference)

3   without and with an homogeneous temporal reference



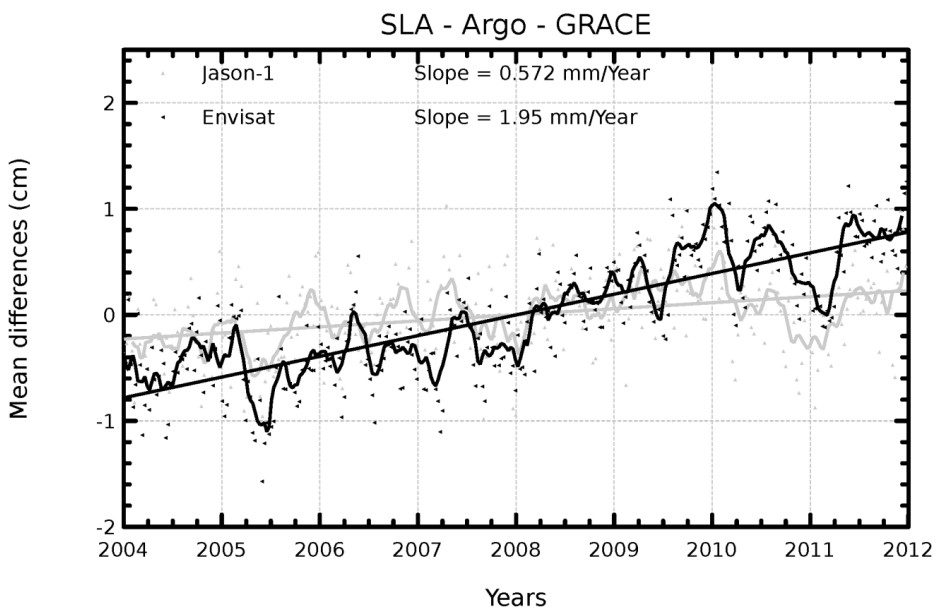

Figure 1 : Mean differences between altimetry and steric + mass contributions from Argo and GRACE

measurements for Jason1 and Envisat missions



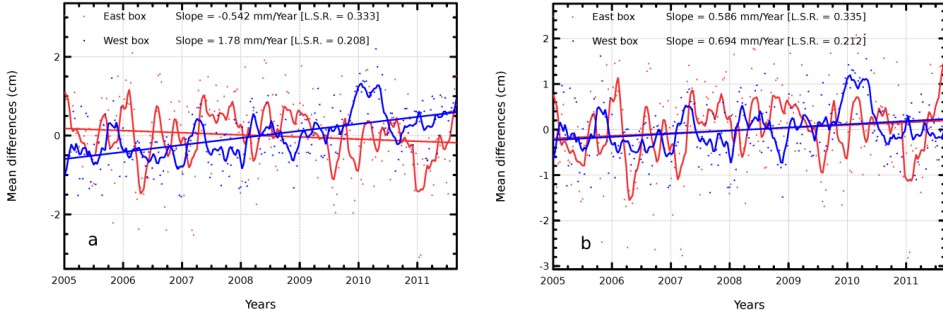

Figure 2 : SSH differences (cm) between Jason-1 altimeter data (cycles 1 to 355) and Argo in-situ measurements
(900 dbar) computed with GDR-D (a) and GDR-E orbit solution (b), separating East box (Lon: 60°/120°, Lat: -
30°/+30°) and West box (Lon: -150°/-190°, Lat: -50°/10°). Corresponding annual and semi-annual signals are
removed. Trends of raw data are indicated and the 2-month filtered signal is added.



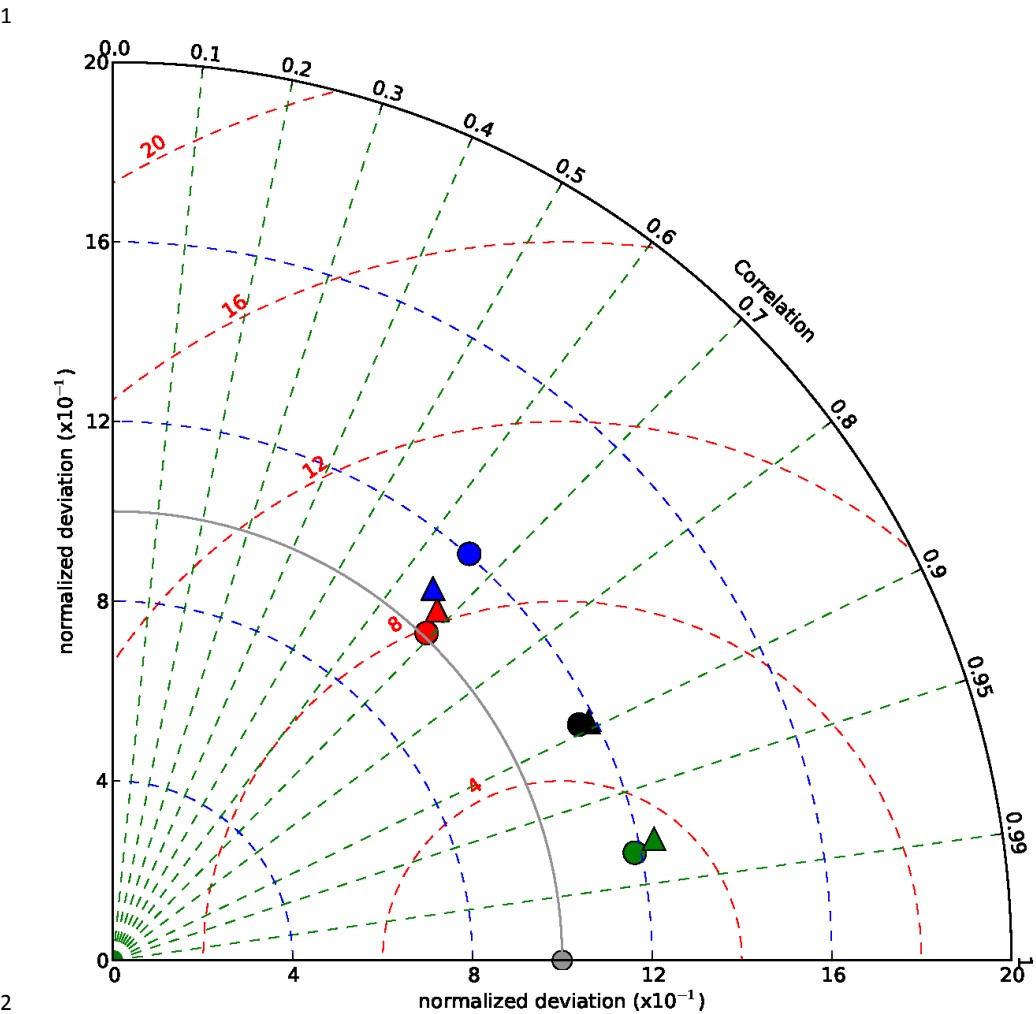

Figure 3 : Taylor diagram of the comparison of CCI (triangles) and AVISO SSALTO/DUACS DT (circles)

merged altimeter sea level products with Argo (900 dbar) and GRACE independent measurements for the global

data (black) and separating high frequencies (red), the annual signal (green) and the inter-annual signals (blue).





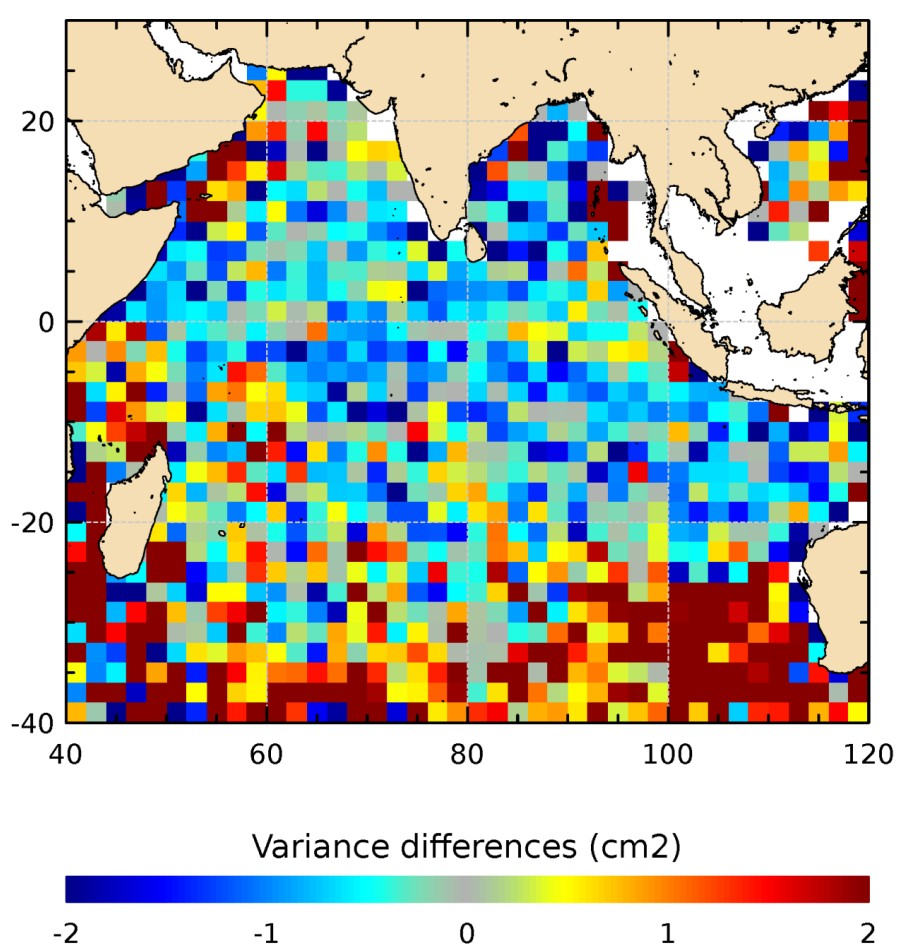

2     Figure 4 : Variance(DUACS 2014-Argo) – Variance(DUACS 2010-Argo) with Argo profiles referenced to 1900

3     dbar over 2005-2012 (cm2)



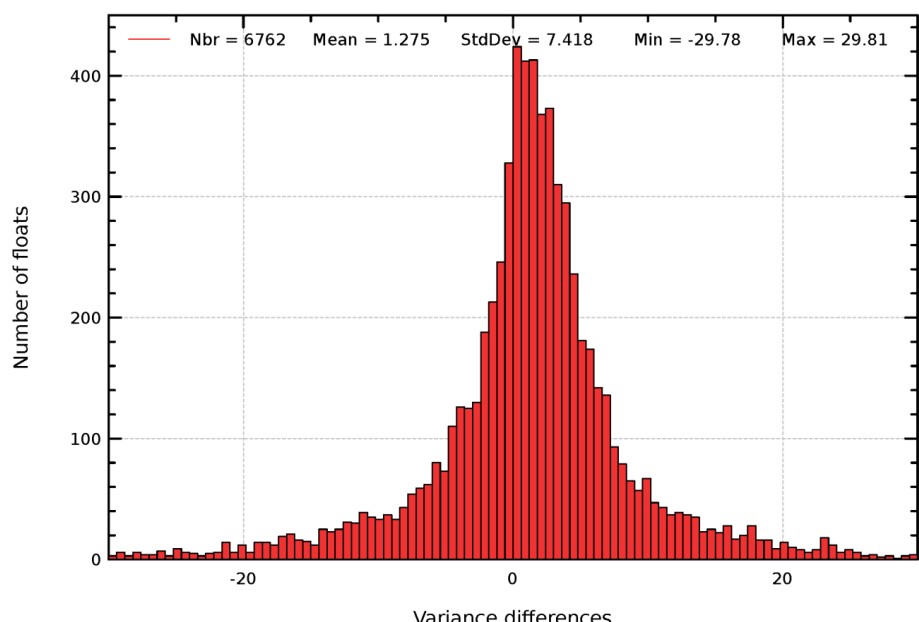

2    Figure 5 : Histogram of the difference of variance of the SLA-DHA differences for each Argo float using

3    successively 1°x1° versus 1°x3° boxes (=Variance(SLA_1x1-DHA) – Variance(SLA_1x3-DHA)) when

4    averaging along-track Jason-1 altimeter SLA before collocating with Argo profiles.





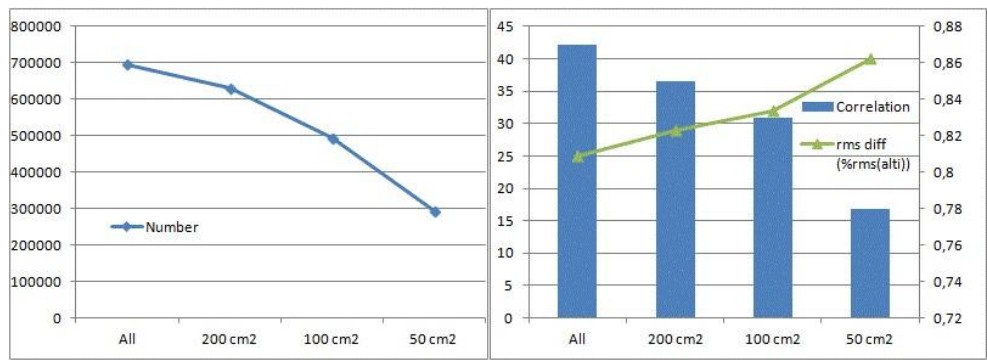

2  Figure 6 : Impact of excluding areas of higher ocean variability than a decreasing threshold: number of observed

3  points (left) and correlation and rms of the differences between AVISO DUACS 2014 and Argo DHA (900 dbar

4  reference) (right).





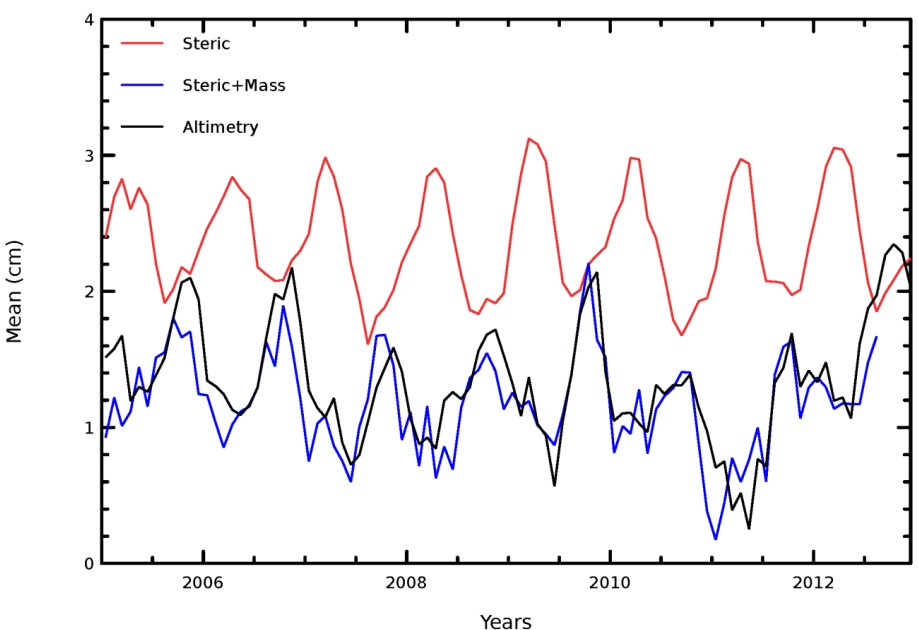

2      Figure 7 : Temporal evolution of the steric DHA from Argo data (red), the summed steric + mass contributions

3      (blue) and the altimeter SLA (black).





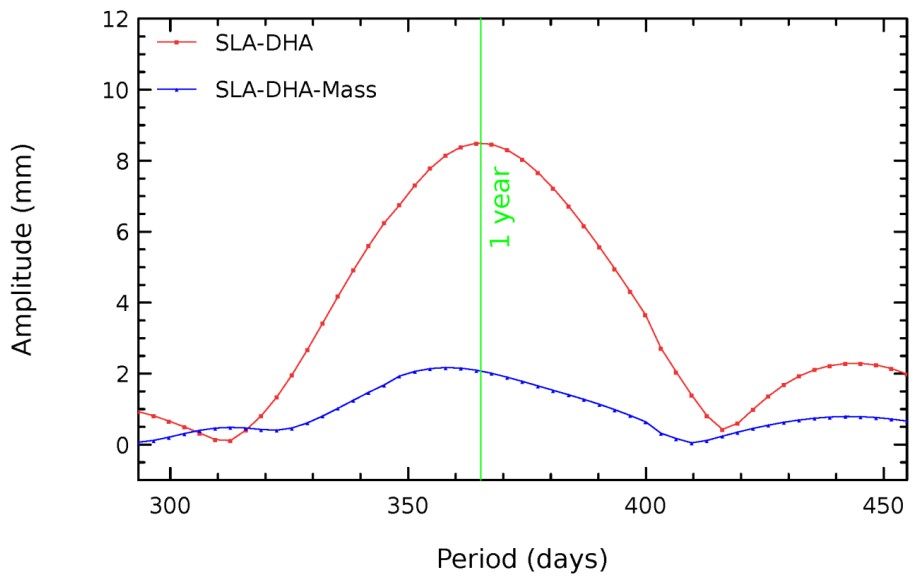

2 Figure 8 : Amplitude of the annual cycle of the differences between Jason-1 altimeter SLA and Argo DHA only

3 (red) or between SLA and DHA + ocean mass (GRACE GRGS V3) (in blue).





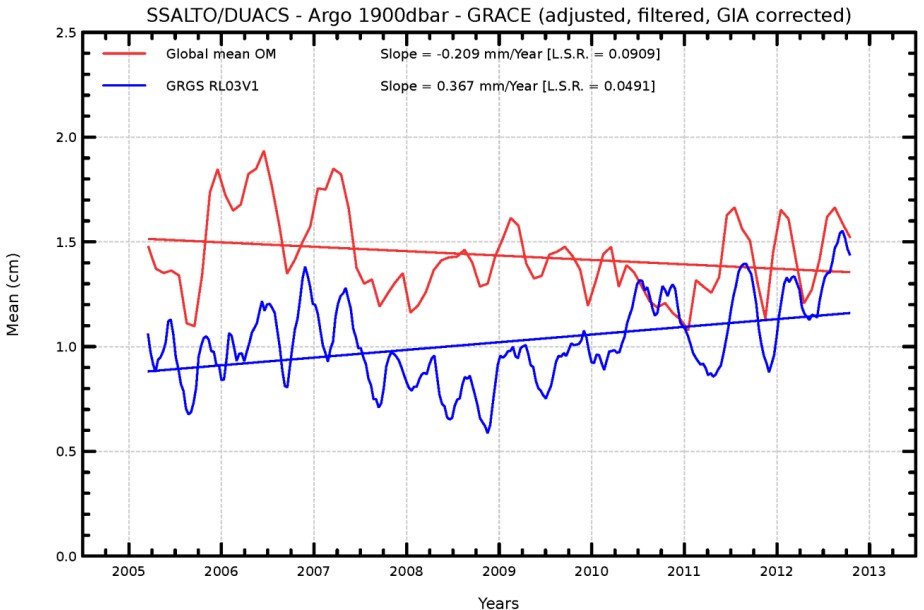

Figure 9 : Differences between SSALTO/DUACS 2014 global MSL and the sum of the Argo steric sea level
(referenced to 1900 dbar) and the GRACE ocean mass contribution derived from the global mean contribution
(Johnson and Chambers, 2013 in red) and the GRGS RL03v1 dataset (Biancale et al., 2014, in blue). Time series
have been adjusted from annual and semi-annual signals, 3-month filtered and corrected from GIA effect. An
arbitrary vertical offset has been applied to the curves for clarity.





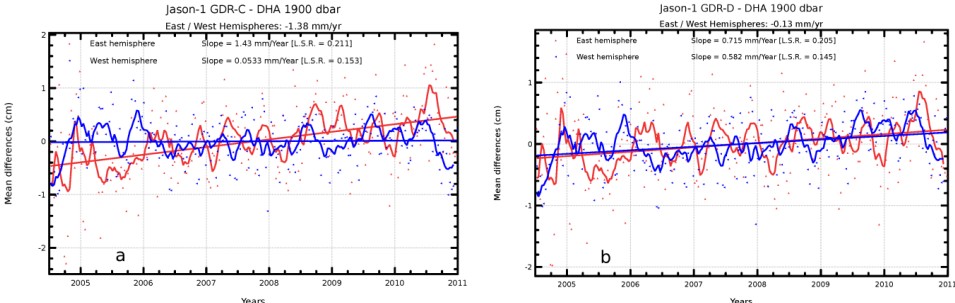

Figure 10 : SSH differences (cm) between Jason-1 altimeter data and Argo (1900dbar) in-situ measurements
computed with GDR-C (a) and CNES preliminary GDR-D orbit solutions (b), separating East (<180°, in red)
and West (>180°, in blue) longitudes. Corresponding annual and semi-annual signals are removed. Trends of raw
data are indicated and the 2-month filtered signal is added.





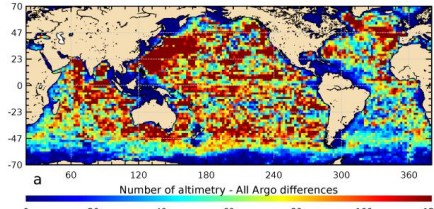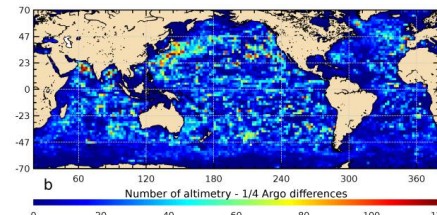

Figure 11 : Number of Argo profiles per 2°x2° boxes over 2005-2012 from all Argo floats (a) and from 25% of

the floats (b).




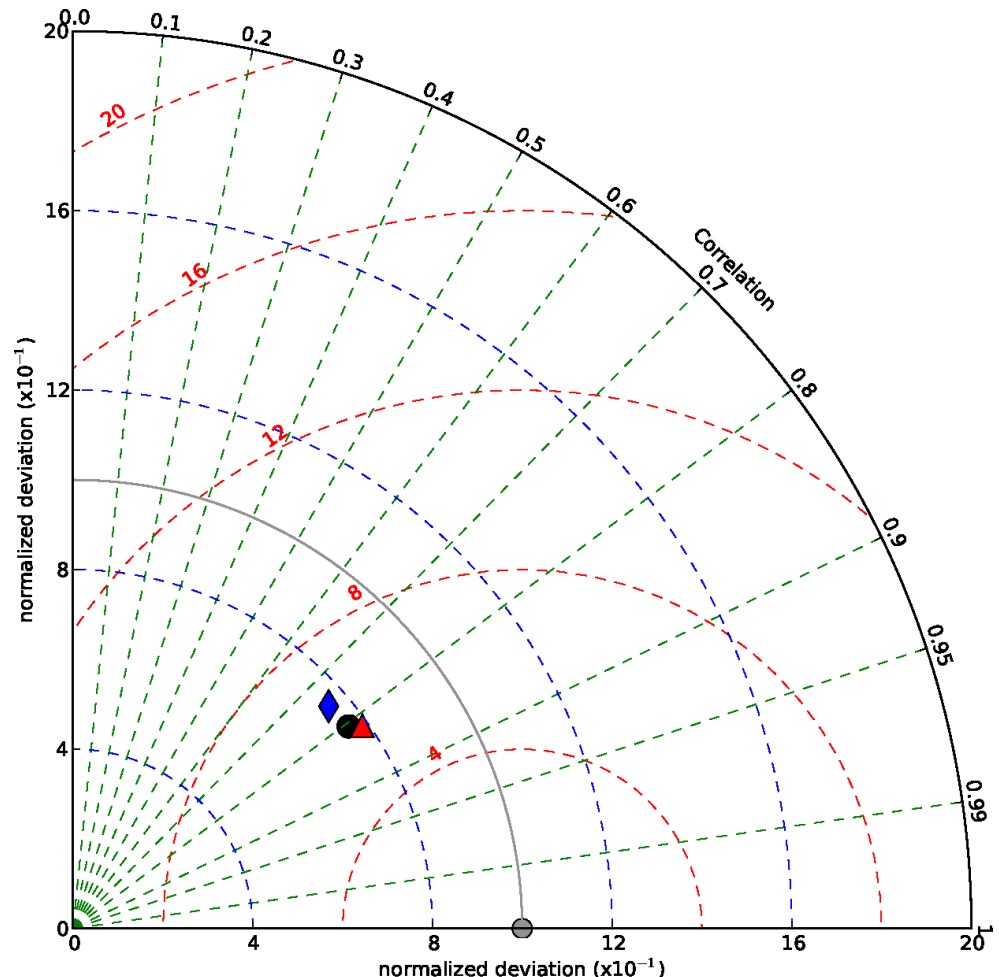

Figure 12 : Taylor diagram of the steric contributions to the sea level derived from different sub sampling of the
Argo floats (DHA referenced to 900 dbar) with the mass contribution (GRACE GRGS) compared with the
AVISO SSALTO/DUACS merged altimeter SLA. For each sub sampling of the in-situ dataset, the
corresponding collocated altimeter measurements are used.





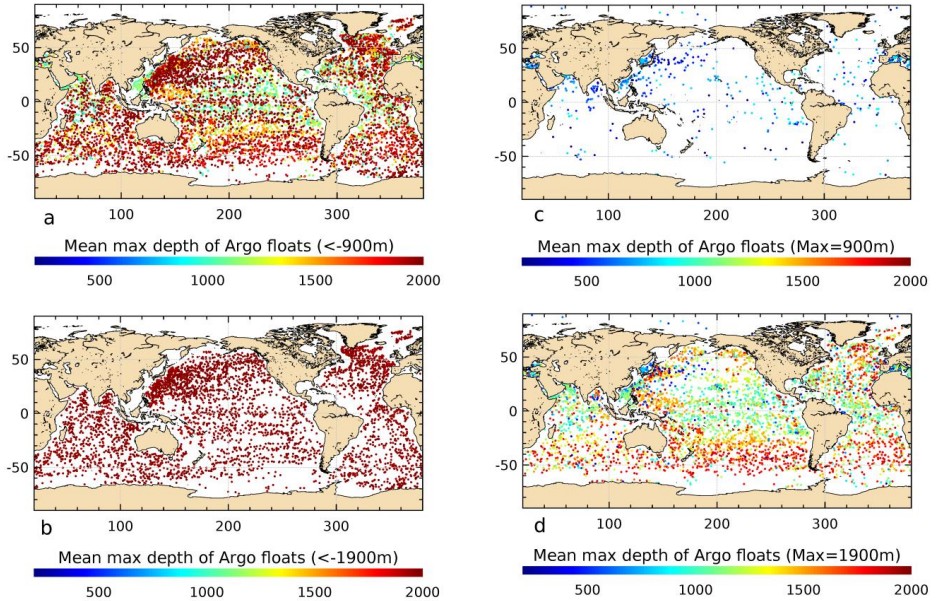

2 Figure 13 : Maps of the mean positions of Argo floats taken into account with a given reference depth (a,b) and

3 the associated floats which will not be used because of their mean max depth shallower than the reference (c,d)

4 for a 900 m (a,c) and a 1900 m (b,d) reference depth over the period 2005-2013.




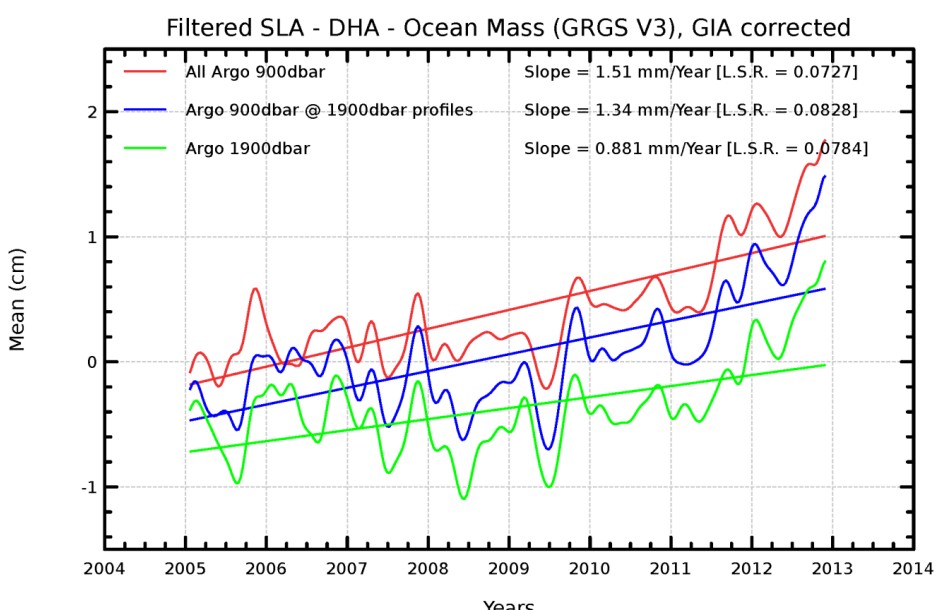

Figure 14 : Global mean sea level trends of the differences between the altimeter mean sea level (AVISO
SSALTO/DUACS 2014) and the steric plus mass (GRACE GRGS RL03) contributions to the sea level with
various subsets of DHA derived from the Argo network: DHA referenced to 900 dbar from all profiles reaching
at least this pressure (red), DHA referenced to 900 dbar from the profiles reaching at least 1900 dbar (blue) and
DHA referenced to 1900 dbar from all profiles reaching at least this pressure (green). All curves are 3-month
low-pass filtered and a GIA correction is applied to altimeter (-0.3 mm/yr) and ocean mass (-1.1 mm/yr)
measurements (Chambers et al., 2010; Tamisiea and Mitrovica, 2011).





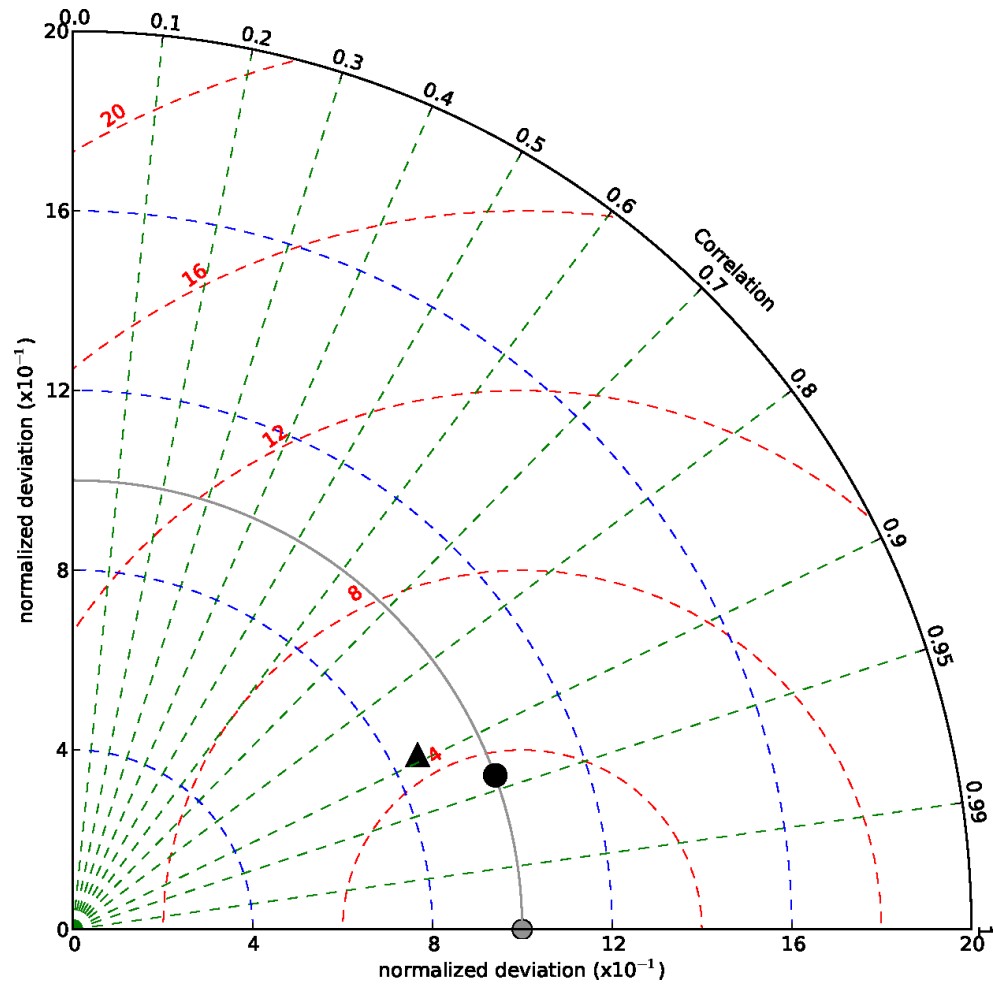

Figure 15 : Taylor diagram of the comparison of the sum of GRACE ocean mass and the steric Argo DHA with a
reference level at 900 dbar (triangle) and 1900 dbar (circle) with altimeter sea level time series
(SSALTO/DUACS 2014) (grey reference circle) on the x-axis over 2005-2013. The blue dotted lines indicate the
normalized standard deviation (altimetry being the reference).



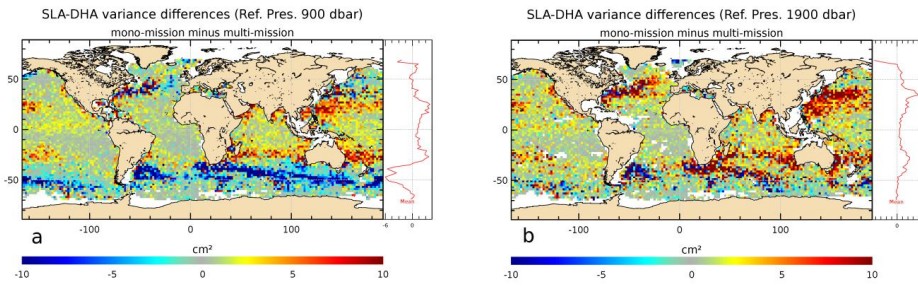

2  Figure 16 : Map of the difference of variance of the altimeter SLA – Argo DHA differences, using successively

3  mono mission and multi missions grids of altimeter products with Argo 900 dbar profiles (a) and 1900 dbar

4  profiles (b).





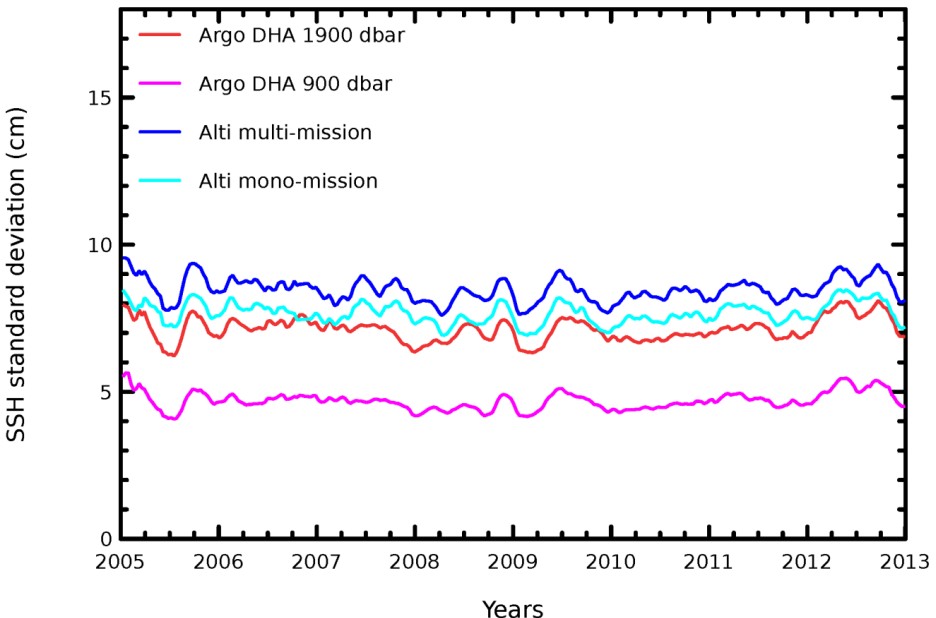

2    Figure 17 : Temporal evolution of the standard deviation of the altimeter SLA derived from mono mission

3    product (light blue), from multi-missions product (dark blue) and from Argo profiles with a 900 dbar reference

4    (magenta) and 1900 dbar reference (red) in the Antarctic Circumpolar Current.





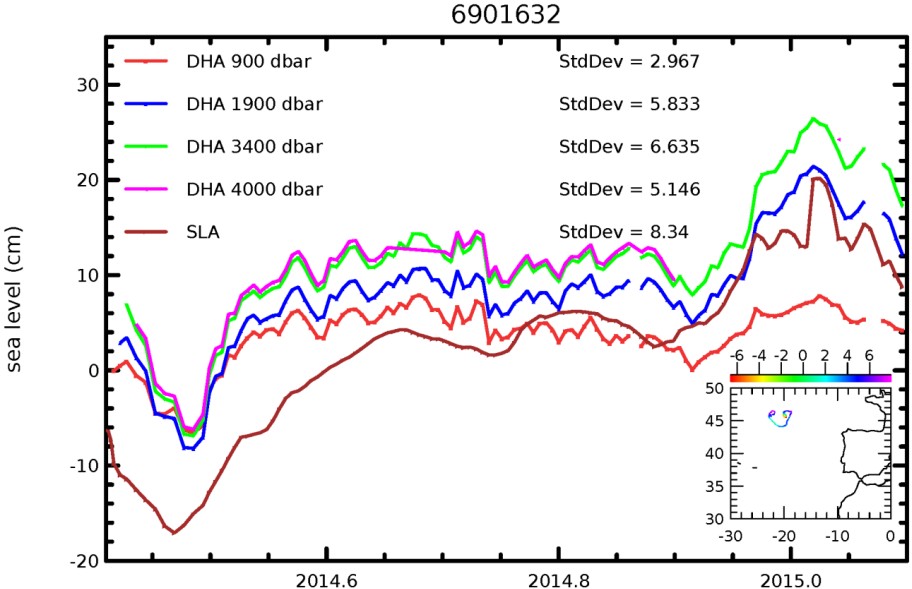

2  Figure 18 : Time series of the DHA derived from the profiles of float WMO 6901632 with different reference

3  levels of integration varying from 900 dbar (red), 1900 dbar (blue), 3400 dbar (green) down to 4000 dbar

4  (magenta) together with the collocated altimeter SLA (brown).