# Peer review of "Analyses of altimetry errors using Argo and GRACE data"

_Ocean Science, 2015_

## Referee Comment (RC1) · D. Chambers (Referee) · 28 Jan 2016

While there is nothing particularly wrong with this manuscript, I suggest it be returned to the authors for a major revision. This is due primarily to the structure of the paper, some issues with the descriptions of the methods and data which make it difficult for a reader to follow what is going on, and some minor English grammar issues that should be reviewed.

My major comments are below, followed by some more minor criticisms.

1. Much of section 3 is confusing, as most of the results are shown later in Section 5. In fact, the authors do a far superior job in Section 5 of: 1) discussing the multiple problems of comparing altimetry and Argo data, and 2) quantifying the uncertainties and errors introduced. Also, many parts of Section 3 are repeated. As such, Section
[Figure]

3 is ultimately redundant. Therefore, I suggest it be deleted, and the last paragraph of Section 1 (Introduction) clearly state the authors are going to quantify the issues with comparing altimetry and Argo data, which has not been done previously (or at least in this detail).

2. Much of section 4 is not particularly new, and just repeats analysis already published by the authors in slightly different forms. I believe they include this here as a motivation for the study, but in my opinion, a brief summary in the Introduction with references to the work (which is already done) is sufficient. Including this section distracts from the important new aspects of the study, which is Section 5.

3. Thus, by deleting Section 3 and 4, and moving Section 5 up to Section 3, the authors introduce the most important aspects of the paper as soon as possible. Any portions of the old Section 3 and 4 that are not already discussed in Section 5 can be included here.

4. Section 2.3 GRACE. I found the explanation of why the mass estimates from GRACE are needed was quite muddled, and I'm an expert and know why. I doubt a novice to the problem reading this would understand. Please rephrase this section to make it clearer. Basically: 1) altimetry measures total sea level, 2) Argo measures upper ocean steric. But there is an additional signal, mass redistribution in the ocean. It's smaller than the steric (except globally), but can be important in some regions. There have been lots of papers dealing with this and can be referenced here [those by Chambers et al., or Ponte et al are good ones], so that a reader can get more detail.

"as proposed by the University of South Florida" would be better as "as provided. . ." or "as estimated by. . .." because proposed means "suggested" and doesn't give the impression it's been calculated.

Also, it's not clear here if the GRGS maps include the time-variable global mean mass or not. The mean mass variation does contribute some to reducing regional residuals. Please clarify this.

5. I have Major concerns with Figure 9 and discussion. "The GRGS dataset has been adjusted for a -1.1 mm/yr GIA effect whereas this effect is already taken into account in the global mean ocean mass time series." The 1.1 mm/year number is a global number and is specific to the averaging kernel used (Chambers et al., 2010). More over, the authors state a few sentences later: "GRGS solution collocated to Argo profiles versus global mean difference) but it is not believed to be a first order contribution to the error"

Thus, the GIA correction WILL be different. I also don't believe sampling GRGS OBP to Argo grids will not create first order difference. I suspect this may explain the inter-annual differences, and the trend. It's easy to check – just do the same calculation with the average of the GRGS grids over the ocean (with a 300 km mask). That should be most comparable to the estimate of Johnson and Chambers. If not, it indicates more problems.

6. Please include fit uncertainty estimates on all trends (and not just formal error – make sure to scale covariance matrix by the variance of the residuals), and tell the reader whether it is standard error, 90% confidence, etc. I understand the authors are waiting until the end to quantify more systematic errors, but when they discuss trends and drifts earlier, the reader needs to know what the fit error is to understand if differences are really meaningful or not.

Minor Comments:

1. Problems with using articles properly in English: For example, here "the" is used too many times. [delete] (add) "Since the early 1990s, several satellite missions have been equipped with altimeters allowing the estimation of Sea Level Anomalies (SLA) and the monitoring of [the] Mean Sea Level (MSL). This contributes to understand(ing) the role of the ocean in the Earth system and to assess the link with [the] global climate change".

One example. The whole manuscript needs a thorough copy-edit by native English speaker.
[Figure]

2. "thermohaline" is misused throughout context. In Oceanography, thermohaline is ONLY used with the vertical circulation related to density differences. While these are caused by temperature and salinity variatiations, it is NOT analogous to expansion/contraction. That's thermosteric/halosteric, or steric for the combined effect. Please change throughout to avoid confusing readers.

3. Section 5.7. I don't quite follow some of this statement. "...the deeper the reference level, the more information from the T/S profiles is taken into account through the water column but the more T/S profiles are not used (those who don't reach the reference level)."

I think what is meant here is that not all Argo floats reach the same level (older ones only went to 1000 dbar). If one selects only the deepest reference floats, one has a reduced number. But using all means a mixture of depths and some loss of sampling of steric variations between 1000 dbar and 2000.

Please reword this so a reader can understand. I think the problem was the authors tried to compress too much info into a single sentence.
* * *

---

## Referee Comment (RC2) · Anonymous Referee #2 · 18 Feb 2016

In this work, the authors present an analysis of altimeter errors by comparing SLA obtained from altimetry with in-situ DHA computed from the Argo array. This independent reference is used to compute the relative altimeter drift and bias. Moreover, GRACE data is used together with the Argo array in order to estimate absolute altimeter drifts. On the other hand, errors associated with the methodology followed to compare the different datasets are investigated through some sensitivity analyses to several parameters related to the datasets.

I think that this paper presents original work and therefore I consider that it must be published in Ocean Science with some minor corrections. I have, however, some suggestions and questions that should be taken into account. There are also some typographical and grammatical errors that need to be corrected.
[Figure]

In the following there is the list of my comments:

[1] My first comment is related to the methodology to compute DHA. In section 3, the authors state that "in-situ DHA are referenced to a synthetic mean Argo Dynamic height calculated over the period 2003 to 2014". This synthetic mean dynamic height was already mentioned in section 2.2 but no information is given concerning to both how this synthetic climatology is obtained and the reference depths used to compute it. I think that a description about the procedure followed to compute it should be included in the text.

[2] In section 4.1 it should be written "... has been related to the altimeter standards ..." instead of "... has been related with the altimeter standards ..." please change it.

[3] Caption of Figure 1 should include the color of the lines associated with Jason 1 and Envisat missions. It is unclear in the figure.

[4] The same for caption of Figure 2. Please include the color related to the East and West boxes.

[5] In section 4.3 the sentence "At low frequencies (in blue), the SL_cci product (triangle) is more in agreement with in-situ data than the SSALTO/DUACS product (circle) which is in favour of a product dedicated to climate studies" is not clear. Please reword it to clarify.

[6] Concerning to correlations given in Tables 1 and 2, it is unclear in the text which is the confidence interval (90%, 95% ...) used to compute them. Please add this information in the new version of the manuscript.

[7] In section 5.2, the authors state that the variability of SLA-DHA is larger in regions of high ocean variability as a consequence of the procedure followed for the collocation of Altimeter (10-days box-averaged along-track SLA) and Argo measurements. In the same paragraph it is mentioned that "this effect could be reduced by computing maps of altimeter measurements by optimal interpolation. However, this is a very time
consuming ..." I wonder if the authors have tried to compute these maps at least for one specific mission in order to check the decrease of the error associated with the collocation of Altimetry and Argo data in these areas of high ocean variability.

[8] Regarding to the previous point, at the beginning of the second paragraph of this section the authors indicate that areas with ocean variability larger than a given threshold are removed before the comparison of altimetry and Argo data; and this fact does not affect the detection of altimeter drifts. Nonetheless, almost at the end of the same paragraph it is written that according to results reported in Figure 6, "This suggests that the areas of large ocean variability significantly contribute to the global statistics computed between altimetry and Argo data". This sentence emphasizes the idea of retaining all the areas of high variability instead of removing some of them before the computation and opposites the aforementioned. I think that this paragraph could be a little bit confusing for the reader and it should be re-written in order to clarify.

[9] Labels and legends in Figure 10 (both panels) should be enlarged in order to make them easier to read.

[10] The first sentence in section 5.7.1 "According to the reference pressure used to integrate the in-situ Argo profiles, no DHA will be computed for all the floats whose mean maximum pressure does not reach this reference level." is a little bit confusing. Please reword it to clarify.

[11] In the sentence in section 5.7.3 "... whereas a reduced variability is found with the in-situ steric measurements referenced to a shallower level (900 dbar) ..." should be specified that it is represented by a triangle in Figure 15 in order to avoid confusion. Moreover, the end of the sentence "... with 0.85 proportion compared with altimeter SLA" should be also re-written to clarify.

---

## Author Comment (AC1) · 12 Mar 2016

We thank Reviewer 1, Don Chambers, for his comments that will be accounted for to improve the manuscript. We respond below point by point to each comment.

[1] Section 3 of the manuscript describes (in less than 20 lines) the method used to compare altimeter and in-situ measurements. We originally thought this part would be useful for the reader. However, we agree that sensitivity analyses associated with all elements of this section are discussed in section 5. So indeed, section 3 is a bit redundant and we agree to delete this section in the next version of the manuscript. We will make sure that enough information is provided in the introduction (section 1) so that the reader is not lost.

[2] Section 4 describes Cal/Val altimetry results achieved thanks to the method of com-

parison. We wanted to present first what the method allows to do, which then leads to the necessity of better characterizing associated uncertainties (section 5). However, we agree that this section 4 looks like a catalog with some references to already published work. So we plan to move most of these results of section 3 in the introduction (section 1).

[3] Following the two previous comments, we will delete sections 3 and 4 and move section 5 up to section 3. Any portions of the old section 3 and 4 that are not already discussed either in the introduction (section 1) or in section 5 will be included here (new section 3).

[4] As requested, we will rephrase section 2.3 in the new version of the manuscript in order to better describe the total and steric sea level and explain why the ocean mass contribution to the sea level is needed. Additional references will be included. As suggested, the sentence "as proposed by the University of South Florida" will be replaced by "as provided by the University of South Florida". Regarding the question whether the GRGS maps include the time-variable global mean mass, we can mention that the estimation of these maps are based on the hypothesis that the total mass of the Earth does not change. Thus, the mean mass over the ocean is varying and it is related with the mass exchange with the continents and the atmosphere. This will be mentioned in the revised version of the paper.

[5] In Fig. 9, the trend of the SLA – DHA – GRGS ocean mass has been estimated after applying a global GIA correction of -1.1 mm/yr to the GRGS ocean mass time series. In addition, the GRGS ocean mass grids have been collocated at the positions and date of each Argo profiles. We don't use the global mean over the ocean. We agree that this global value of GIA correction is specific to the averaging kernel used (Chambers et al., 2010) and is not adapted to the GRGS solution. In order to give an answer to the referee comment (and to compute both curves of Fig. 9 in a more homogeneous way), we have used the GRGS grids over the global ocean with a 300km mask. And (in agreement with the GRGS experts), we have used the mean (over the

global ocean with a 300km mask) of the GIA rates for compressible Earth, using ICE5G ice history and VM2 viscosity profile from F.W.Landerer (Geruo et al., 2013). This leads to a GIA correction of -1.7 mm/yr (instead of the -1.1 mm/yr previously used). Then, the altimeter drift (SLA-DHA-OM) computed with this approach is -0.2 mm/yr, which is the same as the one obtained with the use of the global mean OM from Johnson and Chambers (2013), as in Fig. 9. As both trends are computed homogeneously with this approach (contrary to what was initially done in Fig. 9), we will present these new results in the updated version of the manuscript and stress the importance of the GIA correction.

A. Geruo, J. Wahr, and S. Zhong: Computations of the viscoelastic response of a 3-D compressible Earth to surface loading: an application to Glacial Isostatic Adjustment in Antarctica and Canada, Geophys. J. Int., 2013, 192, 557-572. doi: 10.1093/gji/ggs030

[6] In the paper, the fit uncertainty provided with the trend estimations is what can be called the standard error. This means that the width of the confidence interval of the trend estimations is one standard deviation (33%) of the statistical distribution of the estimator of the trend. Note that in addition of this fit uncertainty, there are systematic errors associated with the method of comparison of altimeter data with in-situ mea-surements. However, when the SLA-DHA differences are computed in two different situations (for instance with a new and a reference altimeter geophysical correction or in the East and West hemispheres), the realizations of these systematic errors are the same in both computations. Thus, they cancel each other, which makes possible to detect some trend differences. These elements will be included in the new version of the article (beginning of the current section 5).

Minor comments: [1] We will ask the editor for a copy-editing of the manuscript by native English speaker. If it is not possible, we will do this before submitting the new version.

[2] The expansion and contraction of the water column due to temperature and salinity

changes will be clearly attributed to the steric effect. We agree that the term "thermo-haline" is rather attributed to the vertical circulation related to density differences and this term will not be used anymore is the context of the manuscript.

[3] The sentence at the beginning of section 5.7 regarding the choice of the reference level of integration of Argo profiles needs indeed some clarification. The comment provided by the reviewer is correct and we propose to mention the following sentence in the updated version: "The integration of the Argo T/S profiles for the computation of the in-situ steric dynamic heights requires a reference level (pressure). As all floats do not reach the same depth, the steric signal will be well sampled through the water column with a deep reference level but the shallower floats will not be used. On the opposite, more floats will be used with a shallow reference level but the vertical steric signal will be less sampled."

---

## Author Comment (AC2) · 12 Mar 2016

We thank Reviewer 2 for his comments that will be accounted for to improve the manuscript. We respond below point by point to each comment.

First of all, please note that following the suggestions of referee #1, the structure of the manuscript will be slightly modified. However, the results and conclusions of the study will be unchanged.

In addition, we will ask the editor for a copy-editing of the manuscript by native English speaker in order to improve the typographical and grammar content. If it is not possible, we will do this before submitting the new version.

[1] In-situ DHA are computed using a synthetic climatology. This synthetic climatology

has been computed following Guinehut et al. (2006): In-situ Dynamic heights (DH) are computed from Argo temperature and salinity vertical profiles with a 900 dbar reference level. These DH can be expressed as the sum of the mean field reference (that we want to estimate) and an anomaly (DH = <DH> + DHA). At this step, the in-situ DHA are considered to be equivalent to the altimeter Sea Level Anomalies (SLA) (in terms of anomaly). Thus, the mean dynamic height is computed as <DH> = DH − SLA at each Argo profile. A global homogeneous field is then computed on a 1° horizontal grid using all DH/SLA pairs and through optimal interpolation." This will be included in section 2.2 of the revised manuscript. Ref: Guinehut, S., P.-Y. Le Traon and G. Larnicol, 2006 : What can we learn from Global Altimetry/Hydrography comparisons ?, Geophys. Res. Lett, 33, L10604, doi:10.1029/2005GL025551

[2] This will be changed in the revised manuscript.

[3] The caption of Figure 1 will be modified as required.

[4] The caption of Figure 2 will be modified as required.

[5] The sentence in section 4.3 will be reworded with: "At low frequencies (in blue), the SL_cci product (triangle) is more in agreement with in-situ data than the SSALTO/DUACS product (circle). As the quality of climate products is rather addressed at these low frequencies (inter-annual and long-term evolution of the sea level), this highlights the better relevance of the SL_cci products for climate studies."

[6] The correlations given in table 1 and 2 have been computed with a 95% confidence interval. This information will be added in the captions of tables 1 and 2.

[7] We state that the approach used for the collocation of altimeter (10-days box-averaged along-track SLA) and in-situ data leads to larger variability of the differences in regions of high ocean variability and that this could be reduced by computing maps of altimeter SLA by optimal interpolation (OI). We have compared the global statistics of the SLA-DHA differences obtained with grids of altimeter SLA from Jason-1 mission

computed successively with the classical box-average method and with OI. The global correlation between altimeter and in-situ data over a seven-year period is significantly increased (+0.1) and the rms of the differences is reduced by 3.2 cm when using altimeter grids computed by OI. These significant improvements are directly associated with the SLA averaging technique: for each SLA map computed by OI, temporal correlation scales are used to weight the data whereas all data have the same weight when averaging over a 10-day window. We have not checked further the impact on the decrease of the error in areas of high ocean variability.

[8] In section 5.2, we mention that the estimation of the trend of SLA-DHA is little affected by removing areas of high ocean variability. And a few lines later, we illustrate with Fig. 6 that the global statistics between SLA and DHA are modified when removing these areas. As this may be confusing, the sentence will be modified with the following: "This indicates that contrary to the trend of the SLA-DHA differences which is less sensitive, the global statistics computed between altimetry and Argo data are significantly affected by the areas of large ocean variability."

[9] Labels and legends of both panels in Fig. 10 will be enlarged so that they can be better read.

[10] The sentence regarding the choice of the reference level of integration of Argo floats in the introduction of section 5.7 will be clarified. Thus, the first sentence of section 5.7.1 will be reworded accordingly as requested: "For a given reference level of integration of the vertical density profiles, only the floats reaching at least this level will be used to compute the associated DHA whereas shallower floats will not be included in the calculation."

[11] In section 5.7.3, it will be specified in the text that the in-situ DHA referenced to 900 dbar is represented by the black triangle in Fig. 15. The end of the sentence will be removed and replaced by "This reduced vertical sampling of the water column leads to a decrease of the DHA standard deviation by a 0.85 factor at global scale."

---

## Author Response (AR1)

**OS62015-111: Author's response following the review process**

Dear editorial board,

First, I would like to thank the two reviewers of my manuscript for their comments. I think they will contribute to improve the quality of the paper and to make it easier to read.

Following the open discussion of my manuscript, I have posted Author Comments (AC1 and AC2) to the two Referee comments (RC1 and RC2). I have provided a point by point response to all referee's comments and listed in details what will be modified in the manuscript.

I now provide the updated abstract and manuscript in line with the Author Comments #1 and #2 with a marked-up version showing the changes made (track changes from MS Word). The structure of the document has been simplified (as suggested by referee 1) so that the most important aspects of the paper are introduced as soon as possible.
Two figures have been removed (1 and 4), the content of figure 9 (now figure 7) has been reprocessed (as suggested by referee 1) and the legend and labels of figure 10 (now figure 8) have been made easier to read (as requested by referee 2).

In addition, as suggested by both referees, the manuscript suffers from English grammatical errors and I would like to ask you if it is possible to perform a copy-editing of the manuscript so that its quality will be improved?
Let me know if this is possible.

Best regards,
Jean-François Legeais

[revised manuscript text omitted]